## Performance evaluation of ROMS v3.6 on a commercial cloud

system

Kwangwoog Jung<sup>1</sup>, Yang-Ki Cho<sup>1, 2</sup>, Yong-Jin Tak<sup>1,2</sup>

5 <sup>1</sup>School of Earth and Environmental Science, Seoul National University, Seoul, Korea

<sup>1,2</sup>School of Earth and Environmental Science/Research Institute of Oceanography, Seoul National University, Seoul, Korea

Correspondence to: Yang-Ki Cho (choyk@snu.ac.kr)

### Abstract

Many commercial cloud computing companies provide technologies such as high-performance instances, enhanced networking and remote direct memory access to aid in High Performance Computing (HPC). These new

- 5 features enable us to explore the feasibility of ocean modelling in commercial cloud computing. Many scientists and engineers expect that cloud computing will become mainstream in the near future. Thus, evaluation of the exact performance and features of commercial cloud services for numerical modelling is appropriate. In this study, the performance of the Regional Ocean Modelling System (ROMS) and the High Performance Linpack (HPL) benchmarking software package was evaluated on Amazon Web Services (AWS) for various configurations.
- 10 Through comparison of actual performance data and configuration settings obtained from AWS and laboratory HPC, we conclude that cloud computing is a powerful Information Technology (IT) infrastructure for running and operating numerical ocean modelling with minimal effort. Thus, cloud computing can be a useful tool for ocean scientists that have no available computing resource.

### 15 Keywords: ROMS, HPC, HPL, Cloud computing, AWS, Enhanced networking

### 1. Introduction

Numerical models are widely used to predict and analyse ocean circulation and various physical property changes. Large amounts of computational power are required for numerical experiments to simulate realistic global ocean circulation. However, preparing sufficient computer resources is difficult owing to economic and physical constraints. Even when the Information Technology (IT) infrastructure is sufficient, installing and preparing the ocean model setup is time-consuming. If IT infrastructures were free from maintenance, ocean numerical models may be more easily and widely used. Efficient configuration and utilisation of IT resources is increasingly being demanded in many fields as well as in the ocean modelling society. In order to satisfy this demand, many companies and organisations are considering or utilising public cloud computing services such as

- Amazon Web Services (AWS) and Microsoft Azure. The number of applications for cloud computing has been steadily increasing. Many studies are being conducted to test whether applications and operations can be ported to cloud computing environments without performance or technical issues. In the early days of commercial cloud services, many experiments associated with the operation of climate models in cloud computing environments were conducted. For example, Oesterle et al. (2015) compared the performance, disadvantages, and merits of
- cloud computing and grids for meteorological model application. Montes et al. (2017) ported and tested AWS as an infrastructure for the Berkeley Open Infrastructure for Network Computing (BOINC) system. Chen et al. (2017) reported that communication latency was an issue for the Community Earth System Model (CEMS) on AWS and parallel speedup remained virtually unchanged when more than 64 cores were used.

Cloud computing is a computing resource utilisation method in which IT infrastructure resources are provided

- through the internet, with fees paid according to computing amount and time of usage. Cloud computing allows researchers, research institutes, and numerical ocean model scientists with limited infrastructure resources such as servers, storage, and electricity to use numerical ocean models at optimal cost without physical difficulties. Three-dimensional numerical ocean models capable of large-scale processing are executed in High Performance Computing (HPC) environments with many cores and Software (S/W) systems such as Message Passing Interface
- (MPI) to increase computation power. In order to execute large-scale numerical models in parallel, parallel systems such as MPI should be implemented properly as well as the configuration of high-speed Network (N/W) devices such as InfiniBand for communication among servers. Expensive Hardware (H/W) and N/W are usually managed by IT professional organisations and engineers. Various studies have been conducted on parallel processing using cloud computing to overcome the problem of high-cost IT infrastructure. However, the cloud
- environment was found to have limitations for parallel processing owing to insufficient functionalities (Oesterle et al., 2015; Chen et al., 2017). Recently, AWS and Azure, which are public cloud computing services, have begun to provide various technological bases such as enhanced N/W and RDMA for effectively implementing HPC. They enable us to easily prepare numerical model environments and conduct numerical experiments anytime and anywhere.
- This study was conducted with the objective of coming up with a method that effectively constructs and executes large-scale three-dimensional numerical ocean models in commercial cloud computing environments with the

latest features such as enhanced N/W and high-performance instances. An additional goal was to also provide a method to improve or extend the performance of such systems in cloud computing environments with real case study data. For this study, the Regional Ocean Modelling System (ROMS), which is a typical community ocean model, was run on AWS. The various performance results and comparison analysis of performance data according

to the node types are presented. We describe how the cluster for the numerical ocean model environment was setup and compare the performance of the numerical model in a commercial cloud computing environment and a laboratory HPC environment.

The remainder of this paper is organised as follows. Section 2 introduces the cloud computing concept and AWS, the commercial cloud computing service used in this study. Section 3 describes the configuration of the ROMS,

the High Performance Linpack (HPL) S/W package, and the experimental conditions for the numerical experiment. Section 4 explains how the ROMS and the HPL S/W package was installed on AWS and in the local laboratory environment for performance comparison. Sections 5 and 6 describe various numerical experimental results obtained for the HPL S/W package and ROMS on AWS and compare them with those obtained in the laboratory HPC. Finally, Section 7 concludes this paper.

### 2. Cloud Computing

### 2.1. Cloud computing overview

Cloud computing provides virtual computer resources in resource pools through the internet with rental fees flexibly charged by usage time and resources. Depending on the type of resource provided, it is possible to distinguish among Infrastructure as a Service (IaaS), Platform as a Service (PaaS), and Software as a Service (SaaS) (Figure 1). Because cloud computing services are provided through the internet, it is possible to use various cloud computing services if internet access is possible. Cloud computing can be categorised as public or private depending on the deployment model (Mell and Grance, 2011). IT companies such as Amazon, Microsoft, and IBM provide public cloud services (AWS, Azure, and Bluemix, respectively) commercially. A private cloud is

25 deployed by a company for internal users and purposes. In this study, we used AWS, a public cloud service that can be used with IaaS option for running a numerical ocean model. Virtualisation is a key technology required to provide services such as IaaS. Through virtualisation, physical servers, storages, and N/W resources can be logically segmented and allocated to users, and logically returned when jobs are completed.

Figure 2 shows the hypervisor, a server virtualisation technology that can divide server resources logically. A physical x86 server can be logically separated and assigned as a Virtual Machine (VM) through the hypervisor (cloudacademy, 2015). The virtual servers in public cloud computing are examples of the utilisation of these hypervisor technologies. The AWS servers used in this study are also VMs provided through this virtualisation technology. As the VMs can be copied and stacked in the repository in the form of images, it is possible to recreate the VMs of the same configurations by additionally creating another copy using the VM image. These techniques

provide a useful method to prepare a number of nodes, which is necessary for large-scale numerical model

experiments. It is helpful to researchers who need to setup highly complicated environments for numerical modelling.

### 2.2. Commercial cloud computing services

- Users of public cloud services have increased rapidly for economic or technical reasons. Major commercial public cloud services in the global market include Amazon's AWS, Microsoft's Azure, IBM's Bluemix, and Google's compute cloud service. The most popular public cloud computing service in the market is Amazon's AWS, which has numerous datacentres and provides many services in various countries. In this study, we constructed and ran the environment for the ocean numerical model on AWS. AWS provides PaaS and SaaS, as well as server resources, according to the user's purpose. In addition, an increasing number of earth science organisations such as NASA
- use AWS to store and process earth-related information (Chen et al., 2017). We selected the high-performance VM servers with high-speed N/W to make the cluster configurations and optimise inter-server communication, and also parallelised the ocean numerical model using them. AWS supplied us with suitable IT resources to achieve our goal.

Table 1 (as of March 2017) gives an example of the various server resources provided by AWS (AWS, 2017c). As

- the performance and functions are separated according to server instance, it is possible to combine the required instances according to the purpose of the research. GPU-equipped instances, which are widely used for deep learning and high-speed processing of images, are also available. Expensive IT resources can be used at a reasonable price according to the usage amount. AWS's prices vary according to datacentre. The most economic server can be selected regardless of the distance between user and server. The datacentre and services in Oregon,
- USA were selected for this study. It is also possible to use IT resources at a much lower cost by using spot-instance type resources instead of on-demand type.

High-speed processor, large memory size, and high N/W throughput are essential for large-scale modelling. In this study, we chose the recent c4-type and r4-type instances with AWS 64-bit Linux for our numerical modelling experiment (AWS, 2017a). The c4 and r4 type instances are appropriate for numerical models that use MPI,

because AWS provides them with high bandwidth of 10 G (r4 type, 20 G) and low N/W latency. Whereas setting up the environment for large-scale models in local HPC is time-consuming, setting up using c4 or r4 instances is not. Copying several VMs for model execution reduces the time required for large-scale modelling experiments. We were able to simulate ROMS for 30 days using eight nodes (c4.8xlarge) for only approximately US\$13.

### 30 3. Numerical Model

### 3.1. High Performance Linpack Benchmarking

HPL, an implementation of Linpack Benchmarking, is a useful tool for evaluating the performance of High Performance Computer Clusters (HPCC) (Rajan et al., 2012). It is a benchmarking software package that solves

a random dense linear system in double precision (64 bit) arithmetic on distributed-memory computers such as MPI clusters. Implementation of the Basic Linear Algebra Subprogram (BLAS) is necessary for its operation. HPL evaluates the general performance of both cloud clusters and local clusters, thereby enabling us to estimate the effect of network and configurations of clusters before performing numerical ocean modelling. The value of N governing complexity of tasks varies from 56000 to 125312 according to the number of processors. The

evaluation result of the cluster performance was calculated as FLoating Point operations per second (Flops).

### 3.2. Numerical Ocean Model

ROMS, which is the numerical model used in this study, is a free-surface ocean model with vertically terrainfollowing and horizontally curvilinear coordinates and solves hydrostatic, free-surface primitive equations

- (Shchepetkin and McWilliams, 2005). A third-order upstream advection scheme and the K-Profile Parameterisation scheme (Large et al., 1994) are used for horizontal advection and vertical mixing, respectively. Many ocean scientists use ROMS in a variety of ways to meet their research needs. ROMS comprises very modern and modular code written in F90/F95 and uses C-pre-processing to activate the various physical and numerical options. It has a generic distributed-memory interface that facilitates the use of several message passage protocols.
- Currently, data exchange among nodes is achieved with MPI. However, other protocols such as MPI2 and SHMEM can be used without much effort. Further, the entire input and output data structure of the model is via NetCDF (ROMS, 2015).

The model domain used in this study extends from  $115^{\circ}$ E to  $162^{\circ}$ E and from  $15^{\circ}$ N to  $52^{\circ}$ N, which includes the Yellow Sea, the East China Sea, and the East/Japan Sea (Figure 3). It features  $1/10^{\circ}$  horizontal grid resolution and

- 40 vertical layers. The bottom topography data is based on the Earth Topography five-minute grid (ETOPO5) dataset of the National Geophysical Data Center (Amante and Eakins, 2009). The initial temperature and salinity were obtained from the National Ocean Data Center (NODC) World Ocean Atlas 2009 (WOA09) (Antonov et al., 2009; Locarnini et al., 2009). For the lateral open boundary, the monthly mean temperature, salinity, and velocity from the Simple Ocean Data Assimilation (SODA; Carton and Giese, 2008) for 2010 were applied. The surface
- 25 forcing, which includes daily mean wind, solar radiation, air temperature, sea level pressure, precipitation, and relative humidity, was derived from the ERA-Interim reanalysis data of the European Centre for Medium-Range Weather Forecasts for 2010 (Dee et al., 2011). These data were applied to calculate the surface heat flux with the bulk formulae (Fairall et al., 1996). Tidal forcing of 10 tidal components was provided by TPXO7 (Egbert and Erofeeva, 2002). Freshwater discharges from 12 rivers were also applied in the model (Vörösmarty et al., 1996;
- 30 Wang et al., 2008). Details on the model area are given in Seo et al. (2014).

### 4. Deployment of the Numerical Ocean Model and the HPL package on AWS and the Laboratory Cluster

The same numerical experiments were conducted in the laboratory HPC environment and on AWS to compare the performance of both environments. The laboratory HPC cluster comprises a three-node cluster consisting of Intel

Xeon 2 CPUs (2.6 GHz, 28 cores) per node. The HPC cluster configured in AWS was an eight-node cluster composed of c4.8xlarge instances. The server instance provided by AWS has virtualised CPU with hyperthreads mode enabled instead of a physical CPU. The optimal number of vCPUs per node with this configuration had to be determined first, and then the optimal number of vCPUs extended for model performance. This is because the

5 performance of virtual CPUs with hyperthreads mode enabled may differ from the performance of physical servers with only physical CPU.

A high-speed N/W environment configuration for MPI-based parallel processing is necessary. The laboratory HPC environment is configured as an InfiniBand high-speed network capable of achieving a maximum bandwidth of 40 Gbps with very low latency. AWS HPC can be configured as an environment supporting an Ethernet-based

- high-speed network having a bandwidth of up to 20 Gbps with low latency (Table 1). In order to secure a bandwidth of 10 Gbps or more and minimise latency, a separate placement group should be constructed and configured with Virtual Private Cloud (VPC) in AWS (AWS, 2016a). A placement group is a logical grouping of instances within a single availability zone (AWS, 2016b). Only in the same placement group is Elastic Network Adaptor (ENA) possible (AWS, 2016c), and so the placement group labelled 'MPI\_(10G)\_on\_Enhanced\_NW' in
- Figure 4 was constructed. A VPC, labelled 'ROMS Cluster on AWS' was constructed in the us-west-2 region (Oregon region) and the connection between nodes made with a private Internet Protocol (IP) address. The parallel application Open-MPI was configured and NetCDF installed for the input and output data structure of the model. A compilation environment is optimised for cloud computing with both PGI compiler 16 and GFortran, which is an open source compiler (Table 2).
- Server resources were virtualised and deployed in AWS. Virtualised IT resources are easier to allocate and manage than physical resources, but performance is slower because physical resources are provided through the software layer. Because the N/W resources are provided via virtualisation, the network is slower than the physical N/W environment. The technology applied to improve the speed of such virtualised N/W resources is Single Root I/O Virtualisation (SR-IOV). AWS also adapts this technology to some high-performance instances. AWS provides an
- additional high-speed N/W environment called ENA to support up to 20 Gbps bandwidth in the r4 type and optimised-EBS storage performance and enhanced N/W up to 10 Gbps bandwidth in the c4 type. If the amount of communication between nodes is large or the number of nodes increases, it is possible to configure the environment using the instance type providing these high-performance features and achieve better numerical modelling performance.
- The SR-IOV is a technical approach to device virtualisation that provides higher I/O performance and lower CPU utilisation than traditional virtualised network devices. Enhanced networking provides higher bandwidth, higher packets per second (PPS) performance, and consistently lower latencies among instances (AWS, 2016d). Placement groups are recommended for applications that benefit from low network latency, high network throughput, or both. An instance type that supports enhanced networking was chosen to provide the lowest latency and the highest PPS network performance for our placement group (AWS, 2016d). Many users may be concerned

placement group and VPC functions, and configuring the connection of the nodes with private IP addresses.

### 5. Results

### 5.1 HPL benchmark simulation

5 Figure 5 compares the performance of the AWS cluster and the laboratory HPC cluster. It can be seen that the performance of the laboratory cluster using HPL is slightly higher than that of the AWS cluster. Further, network latency may be smaller than in the AWS cluster, because the laboratory cluster uses an InfiniBand network. The performance of the two clusters increases linearly with the number of cores. This experimental result suggests that there is only marginal difference in the general performance of the two clusters, which enables us to evaluate the performance of the numerical ocean model in both clusters.

### 5.2 Efficiency simulation

The efficiency of the ocean model in the cluster environment was also evaluated. Figure 6 shows the speedup, the efficiency of the ROMS, and the wall-clock running time with three different grid sizes for 3 days. We define speedup S as follows (Pacheco, 2011):

15 
$$S = \frac{T_{serial}}{T_{parallel}}$$

where T<sub>serial</sub> is the wall-clock time of a single task job, and T<sub>parallel</sub> is the wall-clock tine of the same work in parallel.

The efficiency E is defined as follows (Pacheco, 2011):

 $E = \frac{S}{P}$ 

where S is speedup and P is the number of processor. This experimental result shows that in both clusters the execution efficiency of the ocean model increases proportionally with the number of grids.

### 5.3 Ocean model simulation

Figures 7 and 8 show the simulated Sea Surface Temperature (SST) and surface velocity initially and after 30 days run from 1 January 2010, respectively. The Kuroshio Current, which is characterised by warm water and high speed, is well simulated along the Okinawa trough and the eastern coast of Japan. Cold water appears in the Okhotsk Sea, the northern East/Japan Sea, and the coast of the Yellow Sea as a result of the atmospheric cooling and vertical mixing (Seo et al., 2014). Comparison of the models simulated by AWS and the local servers shows that the Root-Mean-Square Error (RMSE) of the SST is 0.0097 °C and the RMSE of u-component and v-component of the velocity is about 0.0005 ms<sup>-1</sup>. This means that the difference between the simulation results from AWS's HPC modelling and local HPC modelling systems is very small.

### 5.4 Comparison of HPC performance

To examine speedup, groups of 16 c4.8xlarge CPUs were used to extend the AWS cluster. The CPUs were added to the cluster (16, 32, 64, 80, 96, 112, 128 cores) in groups of 1, 2, 4, 5, 6, 7, and 8 nodes in the case of AWS. We increased the number of processors (16, 32, 64, 80 cores) step by step in the ocean modelling test in the laboratory

5 HPC, in which one node has 28 physical cores. We conducted the same incremental increase using groups of 16 CPU units in AWS to compare performance under the same conditions.

Figure 9 shows the result of executing the ROMS in the laboratory HPC and AWS HPC environments, respectively. The execution time for both environments followed a similar reduction pattern. However, the processing efficiency gradually decreased. Small or medium HPC environments (100–200 cores) composed of c4.8xlarge instance and as in AWS have similar performance to a local HPC eluctor.

10 instance nodes in AWS have similar performance to a local HPC cluster.

### 6. Analysis of AWS instance performance

### 6.1 Hyperthreads effect

Many nodes are used to facilitate parallel processing in large-scale numerical models. It is necessary to consider the number of servers and correct performance of the servers in parallel processing, because each server has more cores than in the past. Allocation of the optimal vCPUs for each node and optimising the load balance of each node to ensure enhanced performance are important in cloud computing. Hyperthreads are enabled in the CPUs of AWS instances. However, poor knowledge of their configuration might lead to misunderstanding of a vCPU's performance and consideration of it as being similar to a physical CPU's performance, which may lead to underestimation of the AWS instance's performance.

As shown in Figure 10, there is little difference in the performance of 16 cores and 32 cores. Although 32 vCPUs are in one instance (c4.8xlarge), the actual number of physical cores is 16, because each node provided by AWS has the hyperthreads feature enabled. If a single thread uses 100% of the resource of one physical core, the other threads assigned to that core have to wait to use the physical resources, because the hyperthreads feature of an

25 Intel CPU is virtualised, as shown in Figure 11. If one CPU is used at almost 100% usage, such as a MPI job task, the resource available for another thread will be insufficient. Therefore, a resource capacity plan should be prepared based on this understanding because one node shows optimal performance at almost half of the provided vCPUs. Two instances should be allocated to utilise 16 vCPUs per node rather than using one instance with 32 vCPUs for the performance and the output of a server with 32 physical CPUs in AWS.

### 30 6.2 S/W configurations

The speed of parallel processing can vary depending on the configuration and environment of the software even in the same server environment (Chen et al., 2017). The processing time was measured under an environment with

PGI and GFortran compilers, which are widely used for numerical model compilation. Figure 10 shows the performance comparison between the PGI and GFortran compilers according to the number of vCPUs. As there is negligible difference in performance between the two compilers, PGI compiler 16, which has a commercial version and a community version, was deployed in the following experiments.

### 5 **6.3 Instance type and numbers**

Several types of instances can be used according to research purposes. Two or three instance types were optimised to support computation performance, memory size, and high-speed N/W. We compared the processing performance by selecting the c4 and r4 instance types with enhanced N/W support features. The difference in processing performance according to instance type under the same S/W environment is negligible. It is essential

- 10 to select an instance type that is optimised in advance before driving a large-scale numerical model. The instance type which is optimised for numerical modelling is the c4-type instance, which is composed of the highest computation processing CPU (Intel Xeon, 2.9 GHz). Its performance is better by approximately 5% than the r4 type (Intel Xeon, 2.3 GHz). In a cluster environment consisting of multiple nodes, the modelling performance is similar because the amount of communication between the nodes increases with the number of nodes. The C4
- 15 type is suitable for simulating small-scale numerical models for better CPU performance. However, the difference in the simulation among four or more nodes is negligible because of inter-node communication. Figure 12 compares c4-type and r4-type instances as a function of number of cores. Figure 13 shows that the difference between r4.8xlarge (8 nodes) and r4.16xlarge (4 nodes) is negligible. This result shows that between four and eight the number of nodes does not affect the performance of ROMS.

### 20

### 7. Conclusion

In this study, we investigated the feasibility in terms of parallel processing performanc