# Peer review of "Performance evaluation of ROMS v3.6 on a commercial cloud"

_Geoscientific Model Development, 2017_

## Referee Comment (RC1) · Anonymous Referee #1 · 8 Jan 2018

General comments.

This paper presents a short performance comparison between a cloud platform (amazon web services, provisioned using HPC-specific instances) and a locally available compute cluster. To evaluate the performance differences, it uses two metrics:

1. The high performance linpack benchmark (as used for the top500) 2. A single simulation scenario in the regional ocean modelling system (ROMS).

The motivation for such a comparison is strong. As soon as one moves to simulation problems that are larger than fit on a desktop machine, the question arises: "where can I run my code?". Traditionally, the answer has been "a local compute cluster" or "a supercomputer" (depending on problem size and availability). Now, a third option is

potentially attractive, namely to use a cloud provider.

To my mind, there are number of factors that come in to play here:

1. Accessibility. How easy is to access an appropriate level of compute resource? How quickly can I get access? 2. Usability. How easy is it to get a simulation started? What expertise do I need over and above running a simulation on my desktop machine? 3. Cost. How much do I have to pay to run my simulation? 4. Time to solution. How long does it take to get the results I need? This has (at least) two factors: the speed at which the code runs; and the time it takes to get the model set up, compiled, and through any queues.

This paper addresses, the "code speed" part of time to solution (4), and, to some extent accessibility and usability (1 & 2). There is one sentence discussing cost (page 5, line 28). However, I do not feel that the analysis presented in the paper is comprehensive enough to allow readers to make an informed decision on whether cloud services would provide a suitable avenue to address their simulation needs.

My major concerns are with the level of performance evaluation carried out, and the conclusions drawn. In summary, I do not think that the performance data presented in this paper provide enough evidence to back up the many statements made about the relative performance merits of virtualised hardware.

An example from Section 4:

Page 7, line 20:

"Virtualised IT resources are easier to allocate and manage than physical resources, but performance is slower because physical resources are provided through the software layer. Because the N/W resources are provided via virtualisation, the network is slower than the physical N/W environment."

There are two unsubstantiated statements here:

"Virtualised IT resources are easier to allocate and manage than physical resources."

and, in reference to the network:

"Because the network resources are provided via virtualisation, the network is slower than the physical network environment."

I see no evidence in the paper to support either of the claims. The former is rather subjective, but the latter can surely be objectively measured. On a local system, you could demonstrate that the introduction of a virtualisation layer reduces the performance of the network. Or else, you could cite a study that does this comparison.

The analysis of the results is littered with these kind of statements.

I would encourage the authors to read this paper on benchmarking scientific codes:

@InProceedings{HoeflerBelli2015, author = {T. Hoefler and R. Belli}, title = {Scientific Benchmarking of Parallel Computing Systems}, year = 2015, pages = {73:1–73:12}, month = {Nov.}, publisher = {ACM}, note = {Proceedings of the International Conference for High Performance Computing, Networking, Storage and Analysis (SC15)}, location = {Austin, TX, USA}, }

Some (not an exhaustive list) information that is lacking:

- The spec sheet memory bandwidth of the nodes - The achievable memory bandwidth (e.g. using the STREAM benchmark) - The spec sheet floating point performance of the nodes.

HPL usually achieves close to floating point peak on desktop CPUs. e.g., the first system on the top500 list that just uses Broadwell CPUs (https://www.top500.org/system/178925) has a peak flops of 8128TFlops and achieves 7040TFlops for HPL (around 86% efficiency).

Typical low-order finite difference codes and more likely to be limited by the memory bandwidth, so measurement of that would also be of interest.

[Figure]

In section 5.1, discussing the HPL results:

This section does not provide enough information for a reader to attempt to reproduce the data. For example, what size of matrix was inverted? There is some discussion of HPL in section 3.1, but this does not detail the exact setup:

Page 6, line 5: "The value of N governing complexity of tasks varies from 56000 to 125312". What is N? Why are these numbers so specific? Moreover, the statement that HPL is suitable for assessing the effect of network performance is not borne out either by the cited paper (Rajan et al. 2012), or a cursory analysis of the computational complexity of dense inverse computations. For an NxN matrix, HPL globally moves $O(N^2)$ data and does $O(N^3)$ flops. For asymptotically $O(N)$ flops/byte moved. Even with a very slow interconnect, one does not expect to see any real network effects once N is "large enough".

Some particular comments:

There are good *performance models* for HPL. On commodity hardware, it should be straightforward to report the expected performance. Is 100 GFlop/s on 16 cores a good number?

Page 8, Line 6. The statement that "network latency may be smaller than in the AWS cluster" is not really supported by the presented data. As mentioned above, HPL is relatively insensitive to network performance. If you want to make a statement about network latency, then measure it.

If you want to characterise the network performance, then the Intel MPI benchmarks are a good place to start (https://software.intel.com/en-us/articles/intel-mpi-benchmarks/). You can run these on both the AWS instances you have spun up, and the local cluster. These give you direct information about the quality of the interconnect and MPI implementation, rather than attempting to guess from secondary data. For example, see a Supercomputing 2016 poster comparing AWS instances with the TSUBAME-KFC supercomputer (http://sc16.supercomputing.org/sc-archive/tech_poster/poster_files/post267s2-file2.pdf).

Although the two experimental testbeds give approximately the same floating point performance, it is not possible to determine if the effect of virtualisation is introducing a penalty (because the peak performance of the hardware is not reported anywhere). Are these two lines similar because the hardware is approximately the same, and being used with similar efficiency? Or, does the AWS instance in fact have twice as much "spec sheet" performance, but you lose 50% for some reason (or vice versa)?

Moving on to the discussion of the ROMS simulation:

Section 5.2.

This again has little analysis of the data, other than to note that when using higher-resolution simulations, the scaling efficiency is better. But this is not surprising. When we increase the amount of local work (higher resolution), any effects due to serial fractions (Amdahl's law) from bad parallelisation or similar become smaller. An interesting question is how many degrees of freedom (dofs) are used per core. The C-grid staggering for ROMS means that, if I have done my sums correctly, there are about 1.7e6 dofs for each of velocity and pressure on the smallest grid. On 64 cores, this corresponds to around 3e4 dofs/core. For explicit schemes that have not been highly optimised, this is generally seen to be close to a break-even point for parallel scaling. So the results at this scale are not surprising That the AWS cluster stops scaling sooner is almost certainly due to network performance. This should be possible to model if one knew what the actual network latency and bandwidth were (you could measure this!). For an example of how to make these predictions I can recommend Fischer, Heisey, and Min (2015).

@TechReport{Fischer:2015, author = {Paul F. Fischer and Katherine Heisey and Misun Min}, title = {Scaling limits for {PDE}-based simulation}, institution = {Argonne National Laboratory}, year = 2015, number = {ANL/MCS-P5347-0515}, doi = {10.2514/6.2015-

3049} }

Section 5.3

Validation of the model notes that there is a small difference in the results between local and AWS run simulations. Probably this is fine, but one wonders. Was the comparison on the same number of cores in both cases? Does ROMS offer bitwise reproducible results when run on the same number of cores (or only statistically the same)? For example, if the authors were to run multiple times on a single system, would they expect a small spread in the results, or would they expect the same numbers every time? If the latter, it is not obvious why this should not transfer to different systems (if using the same number of cores).

Section 6.1

The discussion of hyperthreading effects is somewhat confusing. It seems like you are just saying "If you want performance equivalent to X physical CPUs, you should provision X physical CPUs (rather than the X/2 physical CPUs you would get with virtualised hyperthreaded CPUs)".

It is unclear from the discussion if allocating two instances each with 16 vCPUs is twice as expensive as one instance with 32 vCPUs. Furthermore, you do not present any evidence of the performance difference between 16 vCPUs and 32 vCPUs (the referenced Figure 10 compares performance between two different compilers).

Section 6.2

This again does not provide enough information to interpret the data. For example, what compiler versions did you use in both cases, what were the compiler flags? Presumably AWS is not special, and you would expect to see the same difference on the local system as well?

Section 6.3

This compares the time to solution between two different instance types. The difference appears to only be in the frequency of the utilised CPUs. The 2.9GHz CPU is 5% faster than the 2.3GHz CPU. An interesting question is, "how much cheaper is the slower CPU". A 5% slowdown seems like it might be worth paying if the cost is much less. But, considering table 1, it is much higher (presumably because more memory is expensive).

Again, assessing the raw data from this experiment is made needlessly difficult. Given the lower cost and higher performance of the c4 instances, it is unclear why the r4 instance was chosen as a comparison.

Some minor comments:

Page 3, line 17. Chen (2017) draw, I think, bad conclusions about the parallel scaling of CESM on AWS. I suspect their local supercomputing resource would have shown the same tail-off (if they had run that far).

Page 5, line 25. It is unclear why setting up the environment for models is time consuming on local HPC services (but not on AWS).

Why does copying VMs reduce the time for large scale experiments?

Page 7, lines 30–32 are copied verbatim from the AWS documentation. Although a citation is provided, it is (to me) not sufficiently clear that these are quotes, rather than description with referenced work. I have not exhaustively checked the rest of the paper for such examples.

Page 12. The code availability section does not reference HPL at all.

Page 15. Most of the instance types listed in this table are not referred to elsewhere in the paper, why are they therefore included?

The listed memory capacity of Table 1 does not tally with Table 2. If I understand the paper correctly, most simulations were carried out with c4.8xlarge instances (which

[Figure]

Table 1 claims supply 60GB RAM). But Table 2 claims that 128GB RAM were available. Which is correct? Again, I have not exhaustively checked details.

---

## Referee Comment (RC2) · Anonymous Referee #2 · 14 Jan 2018

Review of "Performance evaluation of ROMS v3.6 on a commercial cloud system" by Kwangwoog Jung et al.

**1   General Comments**

This paper presents how to run the Regional Ocean Modelling System (ROMS) and the High Performance Linpack (HPL) on Amazon Web Services (AWS) and makes a comparison with an in-house solution (a classical HPC infrastructure)

I think the paper is a very interesting work that could have a good impact on the area of knowledge but, it needs a revision and multiple improvements before publishing can

be recommended:

- English is correct, but I would suggest reviewing all the document to get some word redundancies removed (this will improve general readability), like in P3 l19-20 for the word "computing": *"Cloud computing is a computing resource utilisation method in which IT infrastructure resources are provided through the internet, with fees paid according to computing amount and time of usage."*

- I think a cost comparison can add more information and value to the paper. On P5 l28 it is said: *"We were able to simulate ROMS for 30 days using eight nodes (c4.8xlarge) for only approximately US$13.",* please elaborate this more and compare it with your in-house system (maybe a table could be interesting).

- Was there any kind of data validation of the outputs from AWS vs local HPC cluster? If so, could you please add them to the paper?

- I suggest adding a section on the paper about pros and cons of running ROMS on the cloud vs running it locally.

- Can you please indicate if ROMS is more CPU or memory or network intensive/bound? Can you please relate this to the type of infrastructure and its impact on any possible bottlenecks?

- Can this work be reproduced with other versions of ROMS? If so, please indicate it.

**2 Specific comments:**

- P3, l19: *"Cloud computing provides virtual computer resources in resource pools through the internet with rental fees flexibly charged by usage time and re-*

*sources.".* This is not exact, it is true that Cloud is usually accessed via the Internet, I suggest a more formal definition like ". . . through Broad Network access (like the Internet) . . . " (e.g. "The NIST Definition of Cloud Computing", http://nvlpubs.nist.gov/nistpubs/Legacy/SP/nistspecialpublication800-145.pdf ).

- P4, l18: It should be: *"Cloud computing provides virtual computing resources . . ."*

- P4, l23: I think mentioning Google on this list of public providers. Also, I recommend making a reference, for instance, to Gartner's magic quadrant for cloud infrastructure providers for 2017.

- P4, l26-35: Please make a reference on how Amazon has been using Xen and relate it to this paragraph.

- P5, l6: You say: *"The most popular public cloud computing service in the market is Amazon's AWS"*, please put a reference to refute this.

- P5, l20: Please define "spot-instance".

- P5, l25: *". . . and low N/W latency"*. Please add values on what is understood as low network latency.

- P7, l1: Please add CPU specific model, not only in here but

---

## Author Comment (AC1) · 9 Mar 2018

Dear Anonymous Referee #1,

We are extremely grateful for your valuable and fruitful comments that helped improve our manuscript. The referee's comments have been given in blue, whereas our responses have been written in black.

5   1. Page 7, line 20: "Virtualised IT resources are easier to allocate and manage than physical resources, but performance is slower because physical resources are provided through the software layer. Because the N/W resources are provided via virtualisation, the network is slower than the physical N/W environment." There are two unsubstantiated statements here: "Virtualised IT resources are easier to allocate and manage than physical resources." and, in reference to the network: "Because the network resources are provided via virtualisation, the
10  network is slower than the physical network environment." I see no evidence in the paper to support either of the claims. The former is rather subjective, but the latter can surely be objectively measured. On a local system, you could demonstrate that the introduction of a virtualisation layer reduces the performance of the network. Or else, you could cite a study that does this comparison. The analysis of the results is littered with these kind of statements. I would encourage the authors to read this paper on benchmarking scientific codes:InProceedings{HoeflerBelli2015,
15  author = {T. Hoefler and R. Belli}, title = {ScientificBenchmarking of Parallel Computing Systems}, year = 2015, pages = {73:1–73:12},month = {Nov.}, publisher = {ACM}, note = {Proceedings of the International Conference for High Performance Computing, Networking, Storage and Analysis (SC15)}, location= {Austin, TX, USA}, }

    Answer:

20  Thank you for recommending the reference on benchmarking scientific codes for our research. We already measured the latency of the N/W layer in a virtualised physical server but did not include this in the manuscript. We will add the result in the revised manuscript according to your suggestion. We found that the introduction of a virtualisation layer reduces the performance of the network. The latency of the physical server is smaller than that of the virtualised server, and the difference is approximately 5~7%.

25  Table R1. Test environment for the N/W latency of the virtualised server and the physical server

|  | CPU | Memory | Hypervisor | OS | Compiler |
|---|---|---|---|---|---|
| Physical server | Intel® Xeon® CPU E5-2680 v3 at 2.50 GHz | 128 GB |  | CentOS 6.9 | Intel Compiler 2018 update 1 |
| Virtual server | Intel Core processor Haswell (2.499 GHz) | 120 GB | KVM | CentOS 6.9 | Intel Compiler 2018 update 1 |

The measured network latency of the physical server in the 1G Switch and the value of the network latency are presented below.

30  Table R2. N/W latency and bandwidth of the physical server according to the message size

| Message size | 1 byte | 2 bytes | 4 bytes | 8 bytes | 16 bytes | 32 bytes |
|---|---|---|---|---|---|---|
| Latency (µs) | 171 | 172 | 172 | 170 | 172 | 172 |
| Bandwidth (MB/s) | 2.52 | 4.9 | 9.8 | 18.6 | 36 | 69 |

The measured network latency of the virtualised server in the 1G Switch and the value of the network latency are presented below.

35  Table R3. N/W latency and bandwidth of the virtualised server according to the message size

| Message size | 1 byte | 2 bytes | 4 bytes | 8 bytes | 16 bytes | 32 bytes |
|---|---|---|---|---|---|---|
| Latency (µs) | 183 | 182 | 183 | 184 | 184 | 186 |

| Bandwidth (MB/s) | 1.99 | 3.91 | 7.66 | 14.66 | 27.3 | 39.7 |
|---|---|---|---|---|---|---|

The latency difference between the physical server and the virtualised server is from 10 to 14 µs when the message size is less than 32 bytes in our test environment.

5    We also tested the HPL of the physical server and the virtualised server. The measured value of the HPL is presented below.

Table R4. HPL performance data of the physical server and the virtualised server

| | Physical server (24 core) | Virtualised server (24 core) |
|---|---|---|
| Gflops | 715 | 640 |

10    However, the performance difference between these physical and virtualised resources has recently been gradually decreasing. This study aims to evaluate the performance of ROMS in a virtualised server of the latest cloud computing environment.

*Change in the revised manuscript:*

15    We cited the references on the performance change in the virtualisation resources (Younge et al., 2011, Gupta et al., 2013) in the revised manuscript and included Tables R1–4 representing the latency difference in the Supplementary section.

20    2. Some (not an exhaustive list) information that is lacking:
- The spec sheet memory bandwidth of the nodes - The achievable memory bandwidth (e.g. using the STREAM benchmark) – The spec sheet floating point performance of the nodes. HPL usually achieves close to floating point peak on desktop CPUs.e.g., the first system on the top500 list that just uses Broadwell CPUs (https://www.top500.org/system/178925) has a peak flops of 8128TFlops and achieves 7040TFlops for HPL

25    (around 86% efficiency).
Typical low-order finite difference codes and more likely to be limited by the memory bandwidth, so measurement of that would also be of interest.

Answer:

30    Thank you. The bandwidth performance of the memory is important in evaluating the performance of the HPC. It should be measured in addition to the Flops, which measure the CPU performance. Following your suggestion, we compared the memory performance of C4.x8large of AWS with the local HPC server using the STREAM benchmark (STREAM, 2016). The physical server memory bandwidth is approximately 5–15% bigger than the virtualised server bandwidth (Table R5).

35    *Change in the revised manuscript:*
Table R5 was included in the Supplementary section.

Table R5. Memory bandwidth of the physical server and the virtualised server using the STREAM benchmark

| Function | Local HPC (MB/s) | C4.8xlarge (MB/s) |
|---|---|---|
| Copy | 79091 | 68431 |
| Scale | 74515 | 68259 |
| Add | 80834 | 77424 |
| Triad | 83858 | 77657 |

3. In section 5.1, discussing the HPL results:

This section does not provide enough information for a reader to attempt to reproduce the data. For example, what size of matrix was inverted? There is some discussion of HPL in section 3.1, but this does not detail the exact setup: Page6, line 5: "The value of N governing complexity of tasks varies from 56000 to 125312". What is N? Why are these numbers so specific? Moreover, the statement that HPL is suitable for assessing the effect of network performance is not borne out either by the cited paper (Rajan et al. 2012), or a cursory analysis of the computational complexity of dense inverse computations. For an NxN matrix, HPL globally moves O(N^2) data and does O(N^3) flops. For asymptotically O(N) flops/byte moved. Even with a very slow interconnect, one does not expect to see any real network effects once N is "large enough".

Answer:

Thank you. More information for reproduction will be included in the revised manuscript. We could achieve a better performance of the HPL test with various options and tuning. However, this study mainly focuses on comparing the performance of the ocean modelling system in cloud cluster and local HPC. We employed the HPL to test the cluster configuration and performance before the intensive usage of the ocean modelling system. Rajan et al. (2012) found a slow performance of the inter-process communication in the 1G interconnect switch using the HPL. The HPL was relatively less insensitive in the network configuration but was a useful tool in evaluating the HPC performance by various configurations.

*Change in the revised manuscript:*

We revised page 5 in line 35 as follows:

'The general performance of cloud and local clusters was tested by the HPL before the intensive usage of the numerical ocean modelling system'.

Tables 6 and 7 have been included in the Supplementary section such that readers may reproduce the data.

Table R6 HPL compile environment

| Architecture | PII_CBLAS | Intel_64 |
|---|---|---|
| Compiler | gcc 4.4, openmpi2.0.2 | parallel_studio_xe_2018_update1_cluster_edition |
| Math Library | Atlas | Intel® Math Kernel Library (Intel® MKL) 2018 |

Table R7 Information on the HPL performance test

| Cores | 16 | 32 | 64 | 84 |
|---|---|---|---|---|
| N (problem size) | 56000, 84000 | 56000, 84000 | 56000, 84000, 125312 | 56000, 84000, 125312 |
| NB (block size) | 128 | 128 | 128 | 128 |
| P*Q (process grid) | 2*8, 4*4, 8*2 | 2*16, 4*8, 8*4, 16*2 | 4*16, 8*8, 16*4, | 4*21, 7*12, 12*7, 21*4 |

4. There are good *performance models* for HPL. On commodity hardware, it should be straightforward to report the expected performance. Is 100 GFlop/s on 16 cores a good number?

Answer:

Thank you. This is not an optimised result because the basic architecture option (arch=PII_CBLAS) without optimisation libraries (e.g. Intel Compiler MKL Library) was used to evaluate the performance. Various performances with a Math supporting library have been evaluated to find the optimal performance. Table R8 shows the results. The optimised performance was close to approximately 70% Rpeak flops. The optimised performance of the AWS and the local cluster (Figure R1) indicated a similar change pattern with the unoptimised performance (Figure 5 in the manuscript) according to the number of cores.

Table R8 HPL performance of the clusters

| Cores | Local HPC (GFlops) | AWSC4.8xlarge (GFlops) |
|---|---|---|
| 16 core | 601 | 566 |
| 32 core | 1161 | 973 |
| 64 core | 1870 | 1935 |
| 84 core | 2339 | 2239 |

*Change in the revised manuscript:*

Figure R1 was added and compared with Figure 5 in the revised manuscript.

[Figure]

Figure R1. Performance of the cluster using the Intel Compiler and the MKL Math Library

5. Page 8, Line 6. The statement that "network latency may be smaller than in the AWS cluster" is not really supported by the presented data. As mentioned above, HPL is relatively insensitive to network performance. If you want to make a statement about network latency, then measure it. If you want to characterise the network performance, then the Intel MPI benchmarks are a good place to start (https://software.intel.com/en-us/articles/intel-mpibenchmarks/). You can run these on both the AWS instances you have spun up, and the local cluster. These give you direct information about the quality of the interconnect and MPI implementation, rather than attempting to guess from secondary data. For example, see a Supercomputing 2016 poster comparing AWS in-C4 instances with the TSUBAME-KFC supercomputer.

Answer:

Thank you. The N/W latency with Linux-qperf and Intel-mpibenchmark (intel-mpi-benchmarks, 2017) was measured according to the message size. The latency showed a performance difference between the AWS cluster and the local cluster on the message size.

*Change in the revised manuscript:*

We added Figure R2 to the Supplementary section.

[Figure]

Figure R2. Comparison of the latency between local-HPC and AWS-HPC according to the message size

6. Although the two experimental testbeds give approximately the same floating point performance, it is not possible to determine if the effect of virtualisation is introducing a penalty (because the peak performance of the hardware is not reported anywhere). Are these two lines similar because the hardware is approximately the same, and being used with similar efficiency? Or, does the AWS instance in fact have twice as much "spec sheet" performance, but you lose 50% for some reason (or vice versa)?

Answer:
Thank you. You might have been misled on the performance of the two clusters (Figure 5) because of the insufficient H/W information in the manuscript. More information on H/W has been added to help the readers better understand our study. The CPU configuration of each node in AWS and local HPC (e.g. Threads, Core, Sockets and CPU Clocks) has also been added. Furthermore, we configured both clusters with Intel Compiler and MKL Library to measure the optimised performance. The cluster performance of AWS was similar to that of the local cluster despite the faster CPU, which might be caused by the performance penalty of virtualisation in AWS.

20

Table R9. CPU specification of the local and AWS clusters

|  | Local cluster node | AWS cluster instance (c4x8large) |
|---|---|---|
| Architecture | x86_64 | x86_64 |
| CPU(s) | 28 | 36 |
| On-line CPU(s) list | 0–27 | 0–35 |
| Thread(s) per core | 1 | 2 |
| Core(s) per socket | 14 | 9 |
| Socket(s) | 2 | 2 |

| Vendor ID | Genuine_Intel | Genuine_Intel |
| --- | --- | --- |
| CPU family | 6 | 6 |
| Model name | Intel® Xeon® CPU E5-2697 v3 at 2.6 GHz | Intel® Xeon® CPU E5-2666 v3 at 2.9 GHz |
| CPU MHz | 2599.843 | 3100.012 |
| Hypervisor vendor: | - | Xen |
| L1d cache | 32 K | 32 K |
| L1i cache | 32 K | 32 K |
| L2 cache | 256 K | 256 K |
| L3 cache | 35840 K | 25600 K |

*Change in the revised manuscript:*

Table R9 has been included in the revised manuscript.

7. Section 5.2. This again has little analysis of the data, other than to note that when using higher resolution simulations, the scaling efficiency is better. But this is not surprising. When we increase the amount of local work (higher resolution), any effects due to serial fractions (Amdahl's law) from bad parallelisation or similar become smaller. An interesting question is how many degrees of freedom (dofs) are used per core. The C-grid

10   staggering for ROMS means that, if I have done my sums correctly, there are about 1.7e6 dofs for each of velocity and pressure on the smallest grid. On 64 cores, this corresponds to around 3e4 dofs/core. For explicit schemes that have not been highly optimised, this is generally seen to be close to a break-even point for parallel scaling. So the results at this scale are not surprising that the AWS cluster stops scaling sooner is almost certainly due to network performance. This should be possible to model if one knew what the actual network

15   latency and bandwidth were (you could measure this!). For an example of how to make these predictions I can recommend Fischer, Heisey, and Min (2015).

Answer:

Thank you. We appreciate your valuable comment. We could measure various networks and band widths

20   according to your suggestion; however, it would be another long story because it needs much time and resource to model based on sufficient data. This is beyond the scope of the present study, which suggests cloud computing as a useful tool for ocean scientists that do not have enough computing resources. We leave further analyses with various configurations, that might be an interesting topic, for the next study.

25

8. Section 5.3. Validation of the model notes that there is a small difference in the results between local and AWS run simulations. Probably this is fine, but one wonders. Was the comparison on the same number of cores in both cases? Does ROMS offer bitwise reproducible results when run on the same number of cores (or only statistically the same)? For example, if the authors were to run multiple times on a single system, would they

30   expect a small spread in the results, or would they expect the same numbers every time? If the latter, it is not obvious why this should not transfer to different systems (if using the same number of cores).

Answer:

Thank you. We compared the simulation results with the same number of cores between AWS and local HPC.

35   For additional information, the root mean square error (RMSE) of the temperature and the velocities between AWS and local HPC were calculated to validate the model results according to the number of cores. The RMSE between AWS and local HP was tiny, regardless of the ROMS version and the number of cores.

Table R10. RMSE of the temperature and the velocities between AWS and HPC according to the number of

40   cores using ROMS v 3.6 (revision 783)

| Cores | Temp | U m/s | V m/s |
|---|---|---|---|
| 16 core | 0.0057 | 4.7613e−004 | 5.0426e−004 |
| 32 core | 0.0097 | 4.9277e−004 | 4.8897e−004 |
| 64 core | 0.0108 | 5.5478e−004 | 5.3697e−004 |

Table R11. RMSE of the temperature and the velocities between AWS and HPC according to the number of cores using ROMS v.3.7 (revision 898)

| Cores | Temp | U m/s | V m/s |
|---|---|---|---|
| 16 core | 0.0112 | 5.2358e−004 | 5.7382e−004 |
| 32 core | 0.0086 | 5.0863e−004 | 5.7747e−004 |
| 64 core | 0.0098 | 5.7419e−004 | 5.9089e−004 |

*Change in the revised manuscript:*

Tables R11 and R12 have been included in the Supplementary section.

9. Section 6.1. The discussion of hyperthreading effects is somewhat confusing. It seems like you are just saying "If you want performance equivalent to X physical CPUs, you should provision X physical CPUs (rather than the X/2 physical CPUs you would get with virtualised hyperthreaded CPUs)". It is unclear from the discussion if allocating two instances each with 16 vCPUs is twice as expensive as one instance with 32 vCPUs. Furthermore, you do not present any evidence of the performance difference between 16 vCPUs and 32 vCPUs (the referenced Figure 10 compares performance between two different compilers).

Answer:
Thank you. As shown in Table R9, the actual number of physical CPUs is half of the vCPU provided by instances. If we turn on the hyperthread feature, we can see the number of vCPUs, which is twice the number of real CPUs. Utilizing the hyperthread function to run several operations is advantageous if the amount of CPU usage is small. The effect is negligible if the physical CPU usage is close to 100% like the actual HPC. We included the result for the 16 core and 32 core ROMS run times using the same compiler on the same node to show this fact (Figure R3).

[Figure]

Figure R3. Comparison of the wall-clock running time of vCores with the same compiler

10. Section 6.2. This again does not provide enough information to interpret the data. For example, what compiler versions did you use in both cases, what were the compiler flags? Presumably AWS is not special, and you would expect to see the same difference on the local system as well?

Answer:
Thank you. We compiled our model using makefile with the same compiler and the same compile option. The compiler configuration for ROMS was included the manuscript. We used the default makefile without optimisation options in both cases. More details on the configuration of the environment in AWS have been included in the revised manuscript.

*Change in the revised manuscript:*

We replaced Table 2 with Table R12 in the revised manuscript

Table R1

| Type | CPU | Memory | Node | OS | Compiler |
|------|-----|--------|------|-----|----------|
| AWS HPC | 32 core (vCPU) Intel Xeon E5-2666- v3 (2.9 GHz) | 60G | 8 | Amazon Linux | PGI Compiler 16.10 NetCDF4 gcc 4.4 Intel Compiler 2018 update 1 |
| Laboratory HPC | 28 core Intel Xeon E5-2697-v3 (2.6 GHz) | 128G | 3 | CentOS 6.9 | PGI Compiler 16.10 NetCDF4 gcc 4.4 Intel Compiler 2018 update 1 |

. 11. Section 6.3. This compares the time to solution between two different instance types. The difference appears to only be in the frequency of the utilised CPUs. The 2.9GHz CPU is 5% faster than the 2.3GHz CPU. An interesting question is, "how much cheaper is the slower CPU". A 5% slowdown seems like it might be worth paying if the cost is much less. Considering table 1, however, it is much higher (presumably because more memory is expensive).
Again, assessing the raw data from this experiment is made needlessly difficult. Given the lower cost and higher performance of the c4 instances, it is unclear why the r4 instance was chosen as a comparison.

Answer:
Thank you very much for your comment. The r4 instance was selected to compare the effect of the N/W performance. The c4 instance provided 10 Gbps of bandwidth using the Intel 82599 virtual function (VF) interface, whereas the r4 instance provided 20 Gbps of bandwidth using the elastic network adapter (ENA) (AWS, 2016). ROMS is relatively less insensitive to N/W because it showed little difference in the performance of c4 and r4.

*Change in the revised manuscript:*

We explained the reason why we chose the r4 instance for comparison in the revised manuscript.

Some minor comments:

12. Page 3, line 17. Chen (2017) draw, I think, bad conclusions about the parallel scaling of CESM on AWS. I suspect their local supercomputing resource would have shown the same tail-off (if they had run that far).

Answer:

Thank you. We agree with you on this point.

13. Page 5, line 25. It is unclear why setting up the environment for models is time consuming on local HPC services (but not on AWS). Why does copying VMs reduce the time for large scale experiments?

Answer:

Thank you for your comment. An easy configuration of the environment is a useful benefit of cloud computing. A large-scale model generally needs a large number of nodes. It takes time to prepare an additional physical environment, such as electric capacity and space. Moreover, the installation of OS and the compiler and the network configuration need additional effort. However, we can easily create a node by cloning a pre-configured image in a cloud system, which is much easier than configuring the environment on a per-node basis in local.

14. Page 7, lines 30–32 are copied verbatim from the AWS documentation. Although a citation is provided, it is (to me) not sufficiently clear that these are quotes, rather than description with referenced work. I have not exhaustively checked the rest of the paper for such examples.

Answer:

Thank you. The AWS documentations we cited were technical function manuals. We thought that copied verbatim was more desirable to prevent misleading, but we revised this part according to your suggestion.

*Change in the revised manuscript:*

The text for the revised manuscript is as follows:

'AWS enhanced networking provides higher bandwidth, higher packets per second (PPS) performance, and lower latencies among instances. AWS recommends placement groups for applications that need low network latency and high network bandwidth (AWS, 2016d)'.

15. Page 12. The code availability section does not reference HPL at all.

Answer:

Thank you. We added the code availability of HPL and STREAM in the revised manuscript.

*Change in the revised manuscript:*

The text on code availability for the revised manuscript is as follows:

**Code availability**

ROMS is publicly available and licensed under the MIT/X License. (Please see the ROMS website at http://myroms.org for details.) Its source code is available for download from the ROMS website via the SVN server. The particular version used for computing the ocean simulations executed in this study is available in trunk, revision 783. How to create and reconstruct the cloud computing infrastructure in AWS and makefile options are explained in Appendix A. The Amazon Machine Image (AMI) is sharable in AWS.

The HPL is publicly available and licensed under the HPL Copyright Notice and Licensing Terms. (Please see the HPL website at http://www.netlib.org/benchmark/hpl/index.html for details.) Its source code is available for

download from the HPL website.

The HPL Copyright Notice and Licensing Terms.

The redistribution and usage in source and binary forms, with or without modification, are permitted provided that the following conditions are met:

5        1. The redistributions of the source code must retain the above copyright notice, this list of conditions, and the following disclaimer.

        2. The redistributions in binary form must reproduce the above copyright notice, this list of conditions, and the following disclaimer in the documentation and/or other materials provided with the distribution.

10        3. All advertising materials mentioning features or use of this software must display the following acknowledgement: This product includes software developed at the University of Tennessee, Knoxville, Innovative Computing Laboratory.

        4. The name of the University, the name of the Laboratory, or the names of its contributors may not be used to endorse or promote products derived from this software without specific written
15         permission.

Disclaimer

THIS SOFTWARE IS PROVIDED BY THE COPYRIGHT HOLDERS AND CONTRIBUTORS 'AS IS' AND ANY EXPRESS OR IMPLIED WARRANTIES, INCLUDING, BUT NOT LIMITED TO, THE IMPLIED WARRANTIES OF MERCHANTABILITY AND FITNESS FOR A PARTICULAR PURPOSE ARE
20 DISCLAIMED. IN NO EVENT SHALL THE UNIVERSITY OR CONTRIBUTORS BE LIABLE FOR ANY DIRECT, INDIRECT, INCIDENTAL, SPECIAL, EXEMPLARY, OR CONSEQUENTIAL DAMAGE (INCLUDING, BUT NOT LIMITED TO, PROCUREMENT OF SUBSTITUTE GOODS OR SERVICES; LOSS OF USE, DATA OR PROFITS; OR BUSINESS INTERRUPTION) HOWEVER CAUSED AND ON ANY THEORY OF LIABILITY, WHETHER IN CONTRACT, STRICT LIABILITY, OR TORT (INCLUDING
25 NEGLIGENCE OR OTHERWISE) ARISING IN ANY WAY OUT OF THE USE OF THIS SOFTWARE, EVEN IF ADVISED OF THE POSSIBILITY OF SUCH DAMAGE.

STREAM is publicly available and licensed under the STREAM Copyright Notice and Licensing Terms. (Please see the STREAM benchmark website at https://www.cs.virginia.edu/stream for details.) Its source code is
30 available for download from the STREAM website.

16. Most of the instance types listed in this table are not referred to elsewhere in the paper, why are they therefore included?

Answer:
35 Thank you. We used the c4 and r4 instances in our study. However, public cloud services and AWS provided various instance types with CPU, memory, and N/W configurations for various purposes and sizes. We introduced various instance types to help users select for their research purposes.

40 17. The listed memory capacity of Table 1 does not tally with Table 2. If I understand the paper correctly, most simulations were carried out with c4.8xlarge instances (which Table 1 claims supply 60GB RAM). But Table 2 claims that 128GB RAM were available. Which is correct? Again, I have not exhaustively checked details.

Answer:
45 Thank you for pointing this out, and we apologise for this typo. 60 GB RAM was corrected appropriately.

**References**

AWS: Enhanced Networking, available at: http://docs.aws.amazon.com/AWSEC2/latest/UserGuide/enhanced-networking.html (last accessed: 03 Apr 2017), 2016d

AWS: AWS Pricing, available at: https://aws.amazon.com/ec2/pricing/on-demand/?nc1=h_ls (last accessed: 03 March 2017), 2017c.

Chen, X., Huang, X., Jiao, C., Flanner, M., Raeker, T., and Palen, B.: Running climate model on a commercial cloud computing environment: A case study using community earth system model (CESM) on Amazon AWS, Computers & Geo., 98, 21–25, doi: http://dx.doi.org/10.1016/j.cageo.2016.09.014, 2017.

Gupta, A., Kale, L.V, Gioachin, F., March, V., Suen, C. H., Lee, B., Faraboschi, P., Kaufmann, R., and Milojicic, D.: The who, what, why and how of high performance computing in the cloud: 2013 IEEE International Conference on Cloud Computing Technology and Science, 306–314, doi:10.1109/CloudCom.2013.47, 2013.

HPL: High-performance Linpack Benchmark, available at: http://www.netlib.org/benchmark/hpl/index.html (last accessed: Feb 2017), 2016.

Intel MPI Benchmark: intel-mpi-benchamrks, available at: https://software.intel.com/en-us/articles/intel-mpi-benchmarks (last accessed: 03 Mar 2018), 2018.

McCalpin, John D., 1995: Memory bandwidth and machine balance in current high performance computers, IEEE Computer Society Technical Committee on Computer Architecture (TCCA) Newsletter, 1995.

McCalpin, John D.: STREAM: Sustainable memory bandwidth in high performance computers, a continually updated technical report (1991–2007) (available at: http://www.cs.virginia.edu/stream)(last accessed: 02 Feb 2018), 2017.

PGI: Community Edition, available at: http://www.pgroup.com/products/community.htm (last accessed: 03 Mar 2017), 2016.

Rajan, A., Joshi, B. K., Rawat, A., Jha, R., and Bhachavat, K.: Analysis of process distribution in HPC cluster using HPL: 2nd IEEE International Conference on Parallel, Distributed and Grid Computing, Solan, India, 85–88, doi:10.1109/PDGC.2012.6449796, 2012.

ROMS: Regional Ocean Modeling System (ROMS), available at: https://www.myroms.org/ (last accessed: 09 May 2017), 2015.

STREAM: Sustainable memory bandwidth in high performance computers, available at: https://www.cs.virginia.edu/stream/ (last accessed: 15 Jan 2018), 2016.

Younge, A. J., Henschel, R., Brown, J. T., Laszwwski, G. V., Qui, J., and Fox, G. C.: Analysis of virtualization technologies for high performance computing environment: 2011 IEEE 4th International Conference on Cloud Computing, 9–16, doi:10.1109/Cloud.2011.29, 2011.

---

## Author Comment (AC2) · 9 Mar 2018

Dear Anonymous Referee #2,

We are extremely grateful for your valuable and fruitful comments that helped improve our manuscript. The referee's comments have been presented in blue, and our responses have been provided in black.

**General Comments**

1. English is correct, but I would suggest reviewing all the document to get some word redundancies removed (this will improve general readability), like in P3 l19-20 for the word "computing": "Cloud computing is a computing resource utilization method in which IT infrastructure resources are provided through the internet, with fees paid according to computing amount and time of usage.

Answer:

Thank you for your comment. We removed the word redundancies in the manuscript to improve general readability. The English language in the revised manuscript has been corrected by native speakers.

2. I think a cost comparison can add more information and value to the paper. On P5 l28 it is said: "We were able to simulate ROMS for 30 days using eight nodes (c4.8xlarge) for only approximately US$13.", please elaborate this more and compare it with your in-house system (maybe a table could be interesting).

Answer:

Thank you. We estimated and compared the cloud computing instance cost and local cluster according to your suggestion (Table R1).

Total price of local cluster (Intel Xeon E5-2967 v3, 28Core, 128GB, 750W) with infini-band switch (40G) was $98,000 in 2015. Assuming that the life span of the local cluster is 3 years, the cluster price per hour would be about $4.3 based on H/W Price, Korean electric charges (KEPCO, 2013). In the cloud cluster, expected cost would be from $3.8 to $12.8 according to service plan (AWS, 2017). The spot-instance type for $3.8 is more economical.

*Change in the revised manuscript:*
We added Table R1 in the Supplementary section.

The text for the revised manuscript is as follows:

'We were able to perform ROMS for 30 days of simulation using eight nodes (c4.8xlarge) for $12.8'.

The following table was also included added in the revised manuscript:

Table R1. Cost comparison of AWS HPC and local HPC cluster for ROMS modeling

| Modeling Environment | AWS HPC cluster | Local HPC cluster |
|---|---|---|
| Model S/W | ROMS v.3.6 | ROMS v.3.6 |
| Grid Size | 422×412×40 | 422×412×40 |
| Nodes | 8 Nodes (Physical:18Core×8) | 3 Nodes (Physical:28Core×3) |
| Execution Time (For 30 days of simulation) | 30 ~ 35 min | 40 min |
| Expected Cost (Spot Instance Type) | $3.8 ($0.47 × 8 Node) | Not applicable |
| Expected Cost (On-demand Type) | $12.8 ($1.58 × 8 Node) | Not applicable |
| Expected Cost (Pre-payment Type (3 Years)) | $5.0 ($0.62 × 8 Node) | $4.3 |
| Expected Cost (Pre-payment Type (5 Years)) | Not applicable | $3.3 |

3.Was there any kind of data validation of the outputs from AWS vs local HPC cluster?   If so, could you please add them to the paper?

Answer:

Thank you for your comment. We calculated the root mean square error (RMSE) for data validation of the outputs from the AWS vs. local HPC cluster.

*Change in the revised manuscript:*
We added the following table showing the RMSEs of the temperature and the velocities between the AWS and local HPC clusters:

Table R2. RMSE of the temperature and the velocities between AWS and HPC according to the number of cores using ROMS v 3.6 (revision 783)

| Cores | Temp | U m/s | V m/s |
|---|---|---|---|
| 16 core | 0.0057 | 4.7613e−004 | 5.0426e−004 |
| 32 core | 0.0097 | 4.9277e−004 | 4.8897e−004 |
| 64 core | 0.0108 | 5.5478e−004 | 5.3697e−004 |

Table R3. RMSE of the temperature and the velocities between AWS and HPC according to the number of cores using ROMS v.3.**7** (revision 898)

| Cores | Temp | U m/s | V m/s |
|---|---|---|---|
| 16 core | 0.0112 | 5.2358e−004 | 5.7382e−004 |
| 32 core | 0.0086 | 5.0863e−004 | 5.7747e−004 |
| 64 core | 0.0098 | 5.7419e−004 | 5.9089e−004 |

4.I suggest adding a section on the paper about pros and cons of running ROMS on the cloud vs running it locally.

Answer:

Thank you for your suggestion. Running numerical ocean modelling in the cloud and local clusters has some pros and cons. We have summarised these pros and cons in Table R4.

*Change in the revised manuscript:*
We included the following table and description in the revised manuscript:

Table R4. Comparison of the pros and cons of running numerical ocean modelling in the cloud and local clusters

| | Cloud cluster | Local cluster |
|---|---|---|
| Accessibility | Easy to access using broad N/W (e.g. internet) | Local network-based access (sometimes remotely) |
| Resource usability | On-demand, spot, reserved instance | Depends on resource scheduling |
| Total cost | Relatively low cost (pay per use) | Relatively high cost because of overhead (management, space, technician, etc.) |
| Management | Online resource management (online control-dash board, self-service) | Offline resource management required(space, cost, human, trouble, etc.) |
| Customizing | Relatively hard to change environment Usable only based on the supported item | Relatively easy to change the local environment |
| Security | Relatively less secure because of remoted space and open network | More secure in local space and closed network |

| Technical support | Relatively slow because of online support | Relatively quick and easy from engineer support |
|---|---|---|

5. Can you please indicate if ROMS is more CPU or memory or network intensive/ bound? Can you please relate this to the type of infrastructure and its impact on any possible bottlenecks?

Answer:

Thank you. We think that ROMS is a more CPU-intensive model rather than N/W intensive. We used the c4.8xlarge instance, which had a similar-performance CPU, using 10G ethernet configuration instead of the infini-band (40G) of the general HPC. Despite the relatively lower performance of the N/W condition, the performance of the ROMS simulation was similar, suggesting that the ROMS is relatively less insensitive to the network.

6. Can this work be reproduced with other versions of ROMS? If so, please indicate it.

Answer:
Thank you for your comment. We tested the running time with other versions of ROMS (version 3.7) according to your suggestion. The running time with the number of cores showed a similar result. The RMSE was also similar (Table R3).

*Change in the revised manuscript:*

We added Table R3 in the revised manuscript.

[Figure]

Figure R1. Comparison of the wall-clock running time for 30 days of simulation according to the cores and

ROMS versions

**Specific comments**

1. P3, l19: "Cloud computing provides virtual computer resources in resource pools through the internet with rental fees flexibly charged by usage time and resources.". This is not exact, it is true that Cloud is usually accessed via the Internet, I suggest a more formal definition like ". . . through Broad Network access (like the Internet) . . . " (e.g. "The NIST Definition of Cloud Computing", http://nvlpubs.nist.gov/nistpubs/Legacy/SP/nistspecialpublication800-145.pdf ).
P4, l18: It should be: "Cloud computing provides virtual computing resources…"

Answer:
Thank you. We corrected this point according to your suggestion.

*Change in the revised manuscript:*

'Cloud computing provides configurable computer resources (e.g. networks, servers, storage, applications, and services) in pools with functions of self-service provisioning and automatic metering of usage and rapid provisioning and users can access through Broad Network access (like the internet)'.

2. P4, l23: I think mentioning Google on this list of public providers. Also, I recommend making a reference, for instance, to Gartner's magic quadrant for cloud infrastructure providers for 2017.

Answer:
Thank you. We agree that Google is one of the numerous public cloud providers.

*Change in the revised manuscript:*
We have included a reference about the cloud provider list in the revised manuscript.

3. P4, l26-35: Please make a reference on how Amazon has been using Xen and relate it to this paragraph.

Answer:
Thank you. AWS configures and optimises a virtual machine with hypervisor technology like Xen. We revised and included a reference according to your suggestion.

*Change in the revised manuscript:*
'Figure 2 shows the hypervisor, a server virtualisation technology that can logically divide server resources. A physical x86 server can be logically separated and assigned as a virtual machine (VM) through the hypervisor. The virtual servers in public cloud computing are examples of the utilisation of these hypervisor technologies. The AWS servers used in this study also optimise the VMs provided through this virtualisation technology like Xen (cloudacademy, 2015). The VMs can be copied and stacked in the repository in the form of images; hence, the VMs of the same configurations can be recreated by additionally creating another copy using the VM image. These techniques provide a useful method to prepare a number of nodes necessary for large-scale numerical model experiments. This is helpful for researchers, who need to set up highly complicated environments for numerical modelling.'

4. P5, l6: You say: "The most popular public cloud computing service in the market is Amazon's AWS", please put a reference to refute this.

Answer:

Thank you. We revised and made a reference following your comment.

*Change in the revised manuscript:*
'One of the popular public cloud services in the market is Amazon's AWS (Gartner, 2017)'.

5. P5, l20: Please define "spot-instance".

Answer:
Thank you. Spot-instance was defined in the revised manuscript.

*Change in the revised manuscript:*
'Amazon EC2 Spot-instances are spare compute capacity in the AWS cloud, which can lower Amazon EC2 cost compared to On-demand prices'.

6. P5, l25: ". . . and low N/W latency". Please add values on what is understood as lower network latency.

Answer:
Thank you for your comment. We tested the network latency of AWS according to the message size.

Table R4. Latency of AWS according to the message size

| Message size | 1 byte | 2 bytes | 4 bytes | 8 bytes | 16 bytes | 32 bytes |
|---|---|---|---|---|---|---|
| Latency (µs) | 36.2 | 38.8 | 36.6 | 35.6 | 40.7 | 36 |

*Change in the revised manuscript:*
'The latency values are between 36 and 41 us when the message size is less than 32 bytes'.

7. P7, l1: Please add CPU specific model, not only in here but.

Answer:

Thank you for your comment. We added a CPU-specific model in the revised manuscript.

*Change in the revised manuscript:*
Tables R5 and R6 showing more information have been added in the revised manuscript.

Table R5. Hardware and software configuration of the AWS and laboratory test environments

| Type | CPU | Memory | Node | OS | Compiler |
|---|---|---|---|---|---|
| AWS HPC | 32 core (vCPU) Intel Xeon E5-2666- v3 (2.9 GHz) | 60G | 8 | Amazon Linux | PGI Compiler 16.10 NetCDF4(4.4.1) Intel Compiler 18 update 1 |
| Laboratory HPC | 28 core Intel Xeon E5-2697-v3 (2.6 GHz) | 128G | 3 | CentOS 6.9 | PGI Compiler 16.10 NetCDF4(4.4.1) Intel Compiler 18 update 1 |

Table R6. CPU specification of the local and AWS clusters

| | Local cluster node | AWS cluster instance (c4x8large) |
|---|---|---|
| Architecture | x86_64 | x86_64 |

| CPU(s) | 28 | | 36 | |
|---|---|---|---|---|
| On-line CPU(s) list | 0–27 | | 0–35 | |
| Thread(s) per core | 1 | | 2 | |
| Core(s) per socket | 14 | | 9 | |
| Socket(s) | 2 | | 2 | |
| Vendor ID | Genuine_Intel | | Genuine_Intel | |
| CPU family | 6 | | 6 | |
| Model name | Intel® Xeon® CPU E5-2697 v3 at 2.6 GHz | Intel® Xeon® CPU E5-2666 v3 at 2.9 GHz | Intel® Xeon® CPU E5-2697 v3 at 2.6 GHz | Intel® Xeon® CPU E5-2666 v3 at 2.9 GHz |
| CPU MHz | 2599.843 | | 3100.012 | |
| Hypervisor vendor: | - | | Xen | |
| L1d cache | 32 K | | 32 K | |
| L1i cache | 32 K | | 32 K | |
| L2 cache | 256 K | | 256 K | |
| L3 cache | 35840 K | | 25600 K | |

**References**

AWS: Spot-Instance, available at: https://aws.amazon.com/ec2/spot/?nc1=h_ls/ (last accessed: 03 Feb 2018), 2018.

AWS: AWS Pricing, available at: https://aws.amazon.com/ec2/pricing/on-demand/?nc1=h_ls (last accessed: 03 March 2017), 2017c.

Cloudacademy: Xen Hypervisor, available at: https://cloudacademy.com/blog/aws-ami-hvm-vs-pv-paravirtual-amazon last accessed: 05 March 2017, 2015.

Gartner: Public Cloud Service, available at: https://www.gartner.com/newsroom/id/3808563 (last accessed: 01 March 2018), 2017.

Mell, P. and Grance, T.: The NIST definition of cloud computing recommendations of the National Institute of Standards and Technology, Special Publication 800–145, NIST, Gaithersburg, available at: http://nvlpubs.nist.gov/nistpubs/Legacy/SP/nistspecialpublication800-145.pdf (last accessed: 10 March 2017), 2011.

ROMS: Regional Ocean Modeling System (ROMS), available at: https://www.myroms.org/ (last accessed: 09 May 2017), 2015.